# Occupational Health Sufferings of Child Waste Workers in South Asia: A Scoping Review

**DOI:** 10.3390/ijerph19148628

**Published:** 2022-07-15

**Authors:** Hasna Hena Sara, Anisur Rahman Bayazid, Zahidul Quayyum

**Affiliations:** BRAC James P Grant School of Public Health, BRAC University, Dhaka 1212, Bangladesh; anisur.bayazid@bracu.ac.bd (A.R.B.); zahidul.quayyum@bracu.ac.bd (Z.Q.)

**Keywords:** child labor, waste worker, occupational health, health suffering, South Asia

## Abstract

Background: Child labor remains a health hazard, affecting the mental, physical, and emotional well-being of children. Children engage in waste management through various channels while constantly working to create a healthier and cleaner environment and exposing themselves to numerous health risks. Thus, this scoping review aims to explore the occupational injuries, health hazards, and sufferings of child waste workers in South Asia. Methods: Following the PRISMA guidelines, a scoping review of available relevant scientific literature was completed to comprehensively analyze the extent of child waste workers’ health suffering. Online databases PubMed, SCOPUS, and Google Scholar were searched for predefined criteria. Collected references were screened with Rayyan web tools and Endnote. Based on study inclusion criteria, a thematic synthesis was performed on the findings of 12 articles. Results: This study’s findings provided deep insights into the most prevalent occupational health sufferings among child waste workers, as depicted in the available literature. Prevalence of injuries like cuts and wounds was found predominant. These injuries are caused by the collection, transportation, dumping, and recycling of waste. Respiratory, musculoskeletal, and skin diseases are more prevalent among child waste worker children than in control groups of the same socioeconomic backgrounds. A higher chance of genetic or neuro-degenerative disorder and DNA mutation indicates a long-term effect on the children working in the waste management sector. Psychological sufferings were the least explored, although very common among child laborers. MPD (Minor Psychiatric Disorder) was very high among waste workers. Regarding healthcare-seeking behavior, traditional methods are preferable rather than formal health facilities. More research is required in this area due to a lack of evidence on the health problems of child waste workers. Conclusions: Occupational hazards were myriad among child waste workers. Though many children are involved in waste management, they are typically excluded from mainstream child protection and support systems, making them more exposed to occupational harassment and injury. Policymakers should design specific programs for these vulnerable groups considering the issues below, i.e., provide protective equipment such as facemasks, gloves, footwear, and rag sorting tools to safeguard them from physical damage and illness, ensure access to health care, to school, and provide basic nutrients to them. Furthermore, the authorities should think of alternative income generating programs for these groups of children.

## 1. Introduction

Child labor is a widespread and persistent global concern as adverse childhood experience obstructs a child’s physical, mental & emotional wellbeing, which may persist throughout life [1]. Despite various initiatives over the last two decades concerning the detrimental impacts of child labor, it continues to be a major challenge. Still, 160 million children are in the labor force, 21.8 million from southern Asia, despite the fact that the International Labor Organization’s (ILO) Minimum Age Convention 138 in 1973 stated that the minimum age for employment “may not be set lower than the age of completion of compulsory schooling and, in any event, not less than 15 years”. Globally, a notable proportion of children engaged in hazardous jobs suffer from acute physical injuries and illnesses [2]. According to the ILO and United Nations Children’s Fund (2021), almost 79 million children are engaged in hazardous work [3]. The ILO also estimates that in 1 year, 106.4 million children aged 5–17 experience a work-related injury; even more suffer illnesses or psychological pain. Hazardous child labor can be observed over a wide range of occupational sectors, including agriculture, construction, mining, manufacturing, domestic service, and waste management [4].

Given that waste management is still poorly managed and often remains unregulated, the disadvantaged in many developing countries, and poor children, find this to be an easy access to livelihood opportunities. Many waste pickers are children and wen in it countries, and they collect waste from households and dispose of this in waste bins and open waste heaps. Thus, they contribute financially to their families’ survival [5]. It is known that waste collectors are exposed to various accidental risks, such as traffic accidents involving waste vehicles, being caught in and between the trash compressor, being cut/punctured by sharp waste materials, slipping, or falling [6,7]. Being a child adds to their difficulties because they are often not fully mature enough to avoid the health risks associated with such work.

Work-related injuries and illnesses are multifaceted public health concerns that have a significant human and financial impact in both developed and developing countries [8,9,10]. Occupational injury refers to any physical injury sustained by a worker in connection with his or her work performance. Since injury is a leading cause of death and disability among children worldwide, preventing child injury is closely connected to other issues related to children’s health. Tackling child injury must be a central part of all initiatives to improve the situation of their health and well-being [11], but hazardous child labor practice has deep and complex roots, so short-term approaches have little influence on bringing any major impact. This cries out for immediate action, yet it is an under-researched area that lacks reliable official data to effectively address the nature and extent of such labor practices. Most importantly, the evidence of adverse health consequences of child labor in the context of Asian countries is limited. Inadequate data is a major issue for children working in hazardous sectors. Insight into the factual situation is necessary to develop a complete and effective policy and framework to eliminate child labor from its roots and to improve children’s overall well-being. Therefore, this study aims to explore and synthesize the evidence on occupational health hazards and sufferings of child waste workers in South Asian countries.

## 2. Materials and Methods

This study utilized a scoping review of the literature and followed the PRISMA guideline [12] to explore the evidence on the occupational health sufferings of child waste workers. These steps included ascertaining journal articles and screening articles, determining the eligibility of articles, and then selecting finally the articles that would be included. This study specifically included a thorough examination of the literature and research that was available on the subjects of occupational hazards and health problems associated with waste management among children.

### 2.1. Eligibility Criteria

Studies were selected according to the criteria outlined below (Table 1).

This study aimed to identify the health problems of child waste workers. In reality, children and adult waste workers work together. Some of the studies might have explored the health problems of waste workers regardless of the age of the workers and could have explored more aspects. In this regard, this review included studies that at least included children below 18 years old age. Studies that include both children and adult populations are also included.

### 2.2. Database Information Sources

PubMed, Scopus and Google Scholar databases were searched using specific search terms. Research articles published between January 2000 and December 2021 were taken into account. The year 2000 was chosen because the topic has not been very widely studied before 2000, as well as to coincide with the adoption of the ILO standards on the Worst Forms of Child Labor in order to ensure the study findings can be generalized under current child labor legislation and guidelines.

### 2.3. Search Strategy

The search terms included but were not limited to those presented in Table 2.

The Boolean search method was adopted to combine search terms in an effort to find literature more relevant to the topics of interest.

### 2.4. Selection of Sources of Evidence

A total of 16,664 search records were identified through three database searches (Google scholar-16537, Pubmed-84, SCOPUS-43). The study’s title, author names, and other reference information were exported and saved in an Excel spreadsheet and also in an Endnote file. After exporting all the searches in Rayyan web tools, 10,215 duplicated items were automatically removed. The remaining 6322 results were screened and articles that were beyond the scope of the study were removed (*n* = 6114). These include book chapters, conference proceedings, or published in languages other than the English language. Among 208 items, 166 were excluded after the abstract screening. After that, 42 were considered for full-text screening while 31 articles were excluded as full text was not found (*n* = 5), the location was wrong (*n* = 1), wrong population (*n* = 20), and wrong outcome (*n* = 5). The additional search included one more article. Finally, 12 research articles were included in this study. Both the first and second authors conducted the initial search. To ensure a reproducible study selection, data charting and critical appraisal processes, the authors followed a calibration exercise. The researchers conducted a meeting after every 3 working days to discuss the search and screening process and develop the data extraction form. Title and abstract screening were conducted by the first and second authors. The third author reviewed the whole process. Search results were saved and managed with Rayyan’s systematic review web tools and Endnote.

### 2.5. Data Charting Process

The authors collaborated to create a data-charting form to choose which variable to extract. In an iterative procedure, two authors charted the data separately, while all three authors reviewed the results and revised the data charting form. A standardized data charting tool was designed for this study which captured the relevant information on key study characteristics. The data from each eligible study was charted by the authors separately. Any disagreements between the authors were resolved through group discussion.

### 2.6. Data Items

Article characteristics, i.e., country of origin, publication year, and study characteristics, i.e., study design, location, author and year of publication, sample size and sampling techniques, outcome variable, occupational health sufferings, physical health sufferings, psychological problems, and healthcare-seeking behavior were abstracted as data items. Data were tabulated in the Excel file and analyzed in an iterative process in discussion with all team members

### 2.7. Critical Appraisal of Individual Sources of Evidence

Assessment of the quality appraisal of included articles was conducted using a modified checklist used in a study in the South Asian context [13]. Four checklists were prepared to check the quality of the studies, all of which consisted of 10 questions. The checklists addressed different study methods, cross-sectional, case-control, qualitative, and mixed-method. The questions were focused on methodology, data collection, and rigorous analysis process. The key questions included “Is the objective clearly defined?”, “Is the sample size adequate?”, “Is the sampling technique random?”, “Is the use of the method justified?” and “Are ethical issues addressed?” Each question was marked 0 to 1, where unsatisfactory processes were scored 0, fully satisfied procedures were marked 1, and partial satisfactory techniques were marked 0.5. Articles with a score of less than 6 are considered low quality, 6–8 as medium quality, and greater than 8 marked as good quality articles. This score gives a clear understanding of the research questions answered, methodological quality, precision, and applicability of the studies. The score or appraisal technique was not used to exclude the articles in the screening process. Among the 12 studies included in this review, only 2 were of good quality, 2 were of low quality, and 8 were of medium quality. Two authors performed the appraisal independently, and no significant differences were recorded. Details are included in Appendix A.

### 2.8. Selection of Sources of Evidence

The whole searching and selection process is presented in the PRISMA flow diagram shown in Figure 1 below:

### 2.9. Characteristics of Sources of Evidence

From 16,664 search results (Google scholar, 16,537; Pubmed, 84; Scopus, 43), 12 articles have been finalized for the scoping review based on the study objective. Among the studies, 8 were quantitative (4 case-control, 4 cross-sectional), 1 qualitative, and 3 were mixed-method studies. One article studied only girls, and one article studied only boys; others included both males and females. Six studies were conducted in India, two in Pakistan, and four in Bangladesh. All 4 studies in Bangladesh covered the issues of the capital city, Dhaka. There was one study from each of the other cities, Islamabad, and Faisalabad in Pakistan, and Patiala, Delhi, Allahabad, Guwahati, Telangana, and Bangalore in India. No studies from Nepal, Bhutan, Sri Lanka, Maldives, and Afghanistan matched the search criteria. The retrieved articles represent most parts of the region, shown in Figure 2.

## 3. Results

This scoping review aimed at documenting the evidence on occupational health hazards and sufferings faced by children engaged in waste management activities. Three broad categories of occupational injuries and suffering were identified through the review, i.e., occupational injuries, physical/health problems, and psychological problems. The study characteristics and methodology for the included articles are shown in Table 3.

12 journal articles met the search criteria regarding occupational injuries and health sufferings of workers related to waste management, and all or most of the respondents and participants were children (aged below 18). The review explored a wide variety of sufferings that the children face. Occupational injuries were discussed in eight of the articles, physical health sufferings in 10 articles, and only six articles mentioned psychological sufferings. One article [20] shed light on indirect sufferings and exposures which are accelerated by working in the waste management sector. Additionally, five of the studies included healthcare-seeking behavior. This has also been included in the review as it is directly associated with health suffering.

About nine of the 12 studies used a semi-structured or structured questionnaire or interview schedule, one study conducted a clinical assessment, and two studies conducted physical examinations. To assess psychological sufferings, two studies used specific scales (one article used SRQ 20, and another article used the Quality-of-life BREF questionnaire and Beck Hopelessness scale). The scales were not validated for use with children. However, there was a difference in considering the cut-off points for SRQ 20 in terms of gender. Cut-off point 8 was considered for boys and 6 was considered for girls [16]. Additionally, interview schedules, FGDs, KIIs, and observations were conducted in six studies.

Among the South Asian countries, only Bangladesh, India, and Pakistan have studies related to the search criteria. Bhutan and Maldives have a very low number of child laborers, and there is no available data on how many children work in the waste management sector. Sri Lanka and Afghanistan have a considerable percentage of child labor (9.2% and 7.5%, respectively), but no data is available on how many are involved in waste management. In Nepal, a large number of children have been involved in rag-picking [26]; unfortunately, no research has been conducted on occupational injuries and health sufferings.

The data show that children working in waste management suffer from skin problems, respiratory problems, gastrointestinal problems, headaches, and body pain more than the general population and are at risk of long-term severe diseases [18,20,23].

Only five articles explored mental and psychological health problems, from 2013 onwards. It is a consideration that the region as a whole had overlooked the health and injuries of child waste management-related laborers in the past. Studies reported that this disruption in mental health in children makes them vulnerable, and they become engaged in illegal work which increases social instability [27]. Minor psychiatric disorders, like stress, anxiety, and depression, affect waste workers significantly [28].

One study conducted a questionnaire survey for child waste workers and FGD for parents and junk dealers, where the prevalence of health problems was reported only for children [22].

### 3.1. Occupational Injuries

According to ILO, an occupational injury is defined as any personal injury, disease, or death resulting from an accident during work; it is therefore distinct from an occupational disease, which is contracted as a result of an exposure over a period of time to risk factors arising from work activity [29]. In this review (Table 4), occupational injuries included cuts (from sharp objects like metals and glass), scratches (from sharp or edgeless objects), burns (from burning wastes), bites (dog bites and snake bites during work, insect bites), muscle and ligament sprains (while carrying waste loads), accidents (road accidents during collection or transportation of wastes or work accidents during handling of garbage), wounds, bruises, contact with poisonous weeds, exposure to chemical fumes, and airborne dust. The review also highlights the importance of using protective gear/equipment in the workplace.

According to these studies, the frequency of being injured was higher among child waste workers, from a minimum of 59.4% in one study [15] to a maximum of 93.3% in another study [16]. This wide range of frequency was observed for multiple reasons. There was a 7 year gap between the two studies that showed the greatest differences [15,16]. They were also rom two different countries. In the later study, which was primarily concerned with MPD and only used injury as an indicator to compare with MPD, the definition of injury was ambiguous [16]. It did not mention he period for which prevalence was considered The earlier study in Dhaka used purposive sampling, and the period considered for prevalence rates was 6 months, which might have had an effect on the comparatively lower injury prevalence rates [15]. The injury rate among rag pickers was 59.4% to 80.3% in Bangladesh, 62.8% in Pakistan, and 70.0% to 93.3% in India. This shows that the overall injury rate was very high in the South Asian region. Six of the studies reported cuts, which ranged up to 84.0%, the highest being in India [14,15,17,21,22,23]. Dog bites and insect bites were also frequent. One article reported that 92.0% of respondents suffered from animal and insect bites [22] and another mentioned that 68.8% of respondents always had to face feces and stray animals [17]. Chemical fumes and airborne dust cause various types of health threats, some of which begin in the place where these are inhaled One article reported on these, and the prevalence of being affected by chemical fumes and airborne dust was 11.2% and 28.0%, respectively [17].

### 3.2. Physical/Health Suffering

Children engaging in waste management correspond to various physical sufferings and diseases. Almost 10 studies explored different types of physical health problems (Table 5).

Findings of the studies indicates that 42.1% to 97.3% of these children had experienced health issues [15,17,21,23,24]. The most common issues were skin problems, respiratory problems, gastrointestinal problems, cold and cough, fever, diarrhea and dysentery, itching, jaundice, headache, and back pain.

Children suffered from a variety of skin problems as a result of their exposure to waste [23]. Skin problems affected 5.7% to 97.3% of those polled [14,15,17,18,23]. The main reason behind this wide range was the definition of skin problems, which varied among the studies. The highest, at 97.3%, was found in one study [23] because it included itching, which is common among different groups of people. About 21.0% to 72.4% of respondents reported body aches, 19.7% to 24.0% reported eczema, 11.1% to 73.3% reported itching, and 41.0% reported fungal infection.

In four studies, 9.2% to 85.3% of respondents had respiratory problems and breathlessness, with the majority suffering at a rate greater than 60.0% [14,17,18,23]. Gastrointestinal and stomach problems were identified in 15.6% to 85.3% of respondents [14,17,18,23]. One study mentioned that 10.0% of children suffered from diarrhea and 30.0% suffered from vomiting [14]. Waste pickers were also prone to jaundice. According to one study, 3.1% of children had jaundice [15].

The prevalence of fever was found to be 2.8% to 85.3% and cough to be 6.4% to 82.7% [14,15,17,18,23]. This wide variation could be attributed to the season in which the study was conducted or to variables considered in the studies. The prevalence of headache and back pain was found to be 3.6% and 6.8% in one study [17], compared to 93.3% and 82.7% in another study [23]. In this case, the variation in prevalence could be due to differences in work and workplace, among other factors. Other diseases investigated and discovered included 51.5% anemia, 14.4% to 52.0% scabies, 10.6% goiter, 6.0% to 53.3% eye problems, 16.0% dental problems, 4.0% to 8.6% worm infestation, 68.0% to 71.4% fatigue, and 88.0% burning sensation [14,15,18,23]. Working in the waste sector hampers children’s physical development, as one study found growth retardation among 64.0% of child waste workers [14].

Many of the children are affected by some sort of physical violence, which includes beating, physical torture, and punishment. Children involved in waste management are also affected by physical acts of violence, along with sexual violence. One study reported that 100.0% of child scavengers experience violence within the home context, 92.0% in the workplace, and 84.0% in communities, whereas girls face sexual violence from their childhood, mainly in the home context, and in the community [25].

Compared to children not involved in waste management or related work, child waste workers suffered 20% more gastrointestinal problems, 30% more skin problems, 40% more respiratory and eye problems, and 47% more aches and pains [23].

Along with these, there were more severe risks studied. One study explored the cell biology of rag pickers and found that cellular changes and cell DNA damage, which cause genetic or neuro-degenerative disorders, atherosclerosis, aging, cell death, DNA mutations, and risk of transformation to malignant cells, was more severe in children exposed to garbage [20]. These cellular changes were estimated to be the result of contamination by heavy metals [30], which are very high in waste and garbage dumping locations [30,31,32]. Studies showed that DNA damage could be the adverse effect of air pollution and smoking [33], or from being exposed to e-wastes [34].

### 3.3. Psychological Sufferings

Acknowledging psychological sufferings is one of the least practiced behaviors in South Asian countries. This results in more psychological damage and is the root cause of many situations of social unrest [35,36].

Due to the socio-economic conditions and nature of their work, waste collectors, rag pickers, and other waste related personnel are among the most badly affected psychologically in the region studied. These psychological problems and sufferings include depression, anxiety, hopelessness, and feelings of insult. The prevalence of these sufferings is presented in Table 6.

Working in the waste sector damages a children’s mental health in many ways. One study in Bangladesh reported that about 68.0% of child waste workers faced developmental/mental retardation [14]. Minor Psychiatric Disorders (MPD) were also discovered among 42.7% of child waste workers in another study, although this study did not validate SRQ-20 for use with children [16]. MPD encompasses both common and minor psychiatric issues such as depression and anxiety. Girls were more affected by MPD than boys, accounting for 53.1% versus 39.8% [16]. One study reported that injured persons, lower-income groups, smokers, pan (betel leaf) consumers, and those who were not satisfied with their job were more affected by MPD [16].

Hopelessness was studied in one study, which discovered that 14.4% of those polled felt hopeless, against 10.0% in the control group [19]. Psychological health hazards were also investigated in a survey, which stated that 18.0% of respondents suffer from these [2]. In other studies, stress, anxiety, and depression were found to be common. One study looked into psychological violence and discovered that 100.0% of child scavengers were victims of psychological violence within the family context and 92.0% in the workplace [25].

### 3.4. Healthcare Seeking Behavior

Although healthcare seeking behavior was not targeted within the research design, it is directly related to health issues and thus included in the study. Five of the 12 studies discussed healthcare seeking behavior (Table 7).

The practice of seeking proper treatment during illness was very poor among child waste workers. Studies found that between 24.0% and 88.0% of children sought any kind of medical treatment while they were ill [14,17,22]. Seeking treatment from medical professionals and hospitals was very low, at 24.0% to 64.2% [17,22] Some studies did not mention professional services, but stated that children sought treatment from traditional practitioners [15]. Batol et al. also reported using ghee/oil and sand/dust as medicine by 28.4% and 13.6% of children [17]. However, seeking healthcare services from pharmacies was notable among the studies, as 40.0% to 44.4% of children used the services of local pharmacies [14,15].

## 4. Discussion

This scoping review explored the myriad health sufferings of children involved in waste management. Child waste workers worked long hours, regardless of whether the was hot or cold, and most did not take breaks. This review found that these children were consistently reported to have a high prevalence of occupational injury. A high prevalence of physical and psychological health suffering among child waste workers was also reported.

Prevalence rates for physical health issues were out of alignment by a wide margin, mostly as reported in two studies—one by Batool et al. (2015) [17], and another by Parvin and Faisal (2005) [23]. While the latter typically displayed a very high frequency of diseases, the former reported a very low frequency. The second study used both point and period approaches, considering six months as the period prevalence, and the first study used prevalence over the previous year. The later study investigated the issue more thoroughly than the earlier one, primarily from an epidemiological perspective. Additionally, it had a smaller sample population (only 75 children in the waste worker group) than the prior study (360 children as waste workers). This could be a contributing factor to the significant disparity in prevalence.

Among the physical health issues, skin problems, respiratory problems, gastrointestinal problems, cold and cough, fever, diarrhea and dysentery, itching, jaundice, headache, and back pain were the most commonly reported physical health problems. Poor hygiene and work practice are mainly responsible for these health problems [14,37,38,39]. Similarly, while being exposed to several hazards on the job, child waste workers lack access to personal protective equipment. The intensity of different health problems and sufferings varies based on the location, as types of waste and the working condition varied accordingly. Most of the waste in residential areas is organic, with a mix of different types of garbage. Methane and hydrogen sulphide, among other gases produced by organic waste, cause major health problems such as headaches, lethargy, etc. In industrial locations, inorganic waste predominates, and is frequently segregated into different categories. Waste workers in these locations are exposed to fewer biologically produced gases, but they are occasionally exposed to chemicals and fumes that cause eye irritation, itching, headaches, and other symptoms. Dermatological problems occur when a person is in contact with hazardous chemicals or from microbes that are abundant in organic waste. Thus skin diseases were more prevalent among child waste workers, and previous evidence also aligns with these findings [40].

This review highlights the most prevalent physical health problems among child waste workers. Due to the types of work, musculoskeletal disorders were very common among children as they were involved in carrying heavy loads, climbing stairs regularly, and working in odd postures. For most of the time, informal child waste workers did not have any designated space to work, thus they did not have the option to sit down or take a rest for a few minutes. This aligns with previous research, i.e., children were in a critical condition because their bodies are more vulnerable to work dangers as they are still maturing. Their skeletons are more susceptible to unusual postures and movements, and their early thermoregulation is more temperature sensitive. All of these covariates put children at a higher risk for musculoskeletal disorders, respiratory problems, and gastrointestinal issues [41,42]. Since child waste workers work in harsh environments (full of dust, fumes and chemical in their surroundings) without any facemask, respiratory problems were more prevalent among. A higher likelihood of respiratory problems was also found in previous research on waste pickers [43,44]. Gastrointestinal problems, including diarrhea, are also reported prevalently by several studies [15,17,18], as in most cases child waste workers have their lunch or snacks beside the waste transfer stations, dustbins, and landfills, or in the worst cases, child waste pickers take food from garbage. This also echoes the findings of research in Dhaka, Bangladesh reporting that the total microbial count in household waste collectors was about 20,000 times higher than in general [45]. The scarcity of clean drinking water also creates abdominal pains and gastrointestinal problems [46,47].

Evidence indicates that child waste workers were particularly prone to injuries due to the inability to distinguish between hazards and threats [48]. Thus, the crucial issues for child waste workers engaged in hazardous employment and working circumstances were emphasized in this review. Broken glass, sharp metals, and pointed objects are primarily (79.2%) responsible for cuts and wounds [17] as child waste workers tend to work with bare hands and feet. This aligns with another systematic review which found that cuts, wounds, and other injuries were very common among waste workers due to poor working conditions [49]. Yet again, child waste workers have to work for a long time, some for more than 10 h per day without any rest [22]. Previous research revealed that child workers who worked long hours with fewer breaks had a high likelihood of injuries [50,51]. Furthermore, data shows that carrying a heavy load, working at night, and being exposed to physical risks during long hours of work increases the risk of occupational injury among children by 40% [42]. Injuries were very common among child waste workers as they tend to work without any protective equipment. These threats can be effectively mitigated by providing safety equipment and raising public knowledge about the importance of keeping dangerous and sharp objects separated. According to research, both the workplace and households can be filthy, and protective gear is “useless” to respondents. As a result, lack of awareness is another major issue that policymakers should consider. Another major concern is that much of the protective gear was designed for adults, therefore children are unable to use it effectively [52]. Although only in the context of this region, addressing all informal laborers and where child labor is banned, or looking at the circumstances that validate child labor would be a very complex task. Still, recruiters and authorities should scrutinize child-user-friendly safety equipment and make it available to everyone, including children, in order to break the stigma and build a habit.

This review further highlights the psychological suffering of waste workers and found that, among the included articles, most tended to discuss physical health suffering. neglecting psychological suffering. These psychological problems and suffering include depression, anxiety, hopelessness, loss of memory, and irritability. The least importance was given to mental health issues nd psychological violence. Only 6 of 12 studies explored the topic, and the findings indicate that it is one of the most common forms of suffering. This also echoes the findings of the estimates of the World Health Organization (WHO) that the most common abuse among children is psychological abuse [53]. This aligns with research findings, where it was estimated that 50.0% of children across the world aged 2–17 years have experienced some form of abuse [1]. Minor Psychiatric Disorders (MPD), which include stress, and anxiety among others, were very high among rag pickers compared with others of the same socioeconomic background, due to hazardous working and living conditions as well as an unorganized social life [16]. This is similar to previous research findings that found that MPD was most frequent among waste pickers [54]. Since waste workers are involved in frequent static body postures, they tend to have musculoskeletal problems [55,56], and all of these trigger anxiety and melancholy. Studies also revealed that this kind of monotonous work has been associated with psychological problems [57]. Again, lower quality of life increases hopelessness [19]. Hopelessness is one of the causes that makes workers more exposed to occupational health hazards. In general, hopelessness is prevalent among older people, but when a child starts to work in these conditions, they become hopeless at an early age. This affects their entire lifestyle and future planning. Although males have more chance of getting injured physically, psychological suffering is more prevalent among females, as in a previous study [54]. In addition to common psychological suffering, any problems a male waste worker must face, a girl or woman has to face even more. The social perspective in the south Asian region can slander women on every occasion and in every place, especially working women who face these problems. In this situation, intriguing factors concerning female suffering can be understood from a feminist approach, where the sufferings is viewed from the sufferer’s perspective [58].

## 5. Strength and Limitation of the Study

To our knowledge, this is the first study to document the occupational health risks faced by child waste workers. However, there are several limitations to this review. The search strategy omitted citation tracking and excluded non-peer-reviewed literature. Due to methodological constraints in the initial investigations of some articles, the conclusions that can be derived are limited. Most cross-sectional studies used convenience or purposive sampling, and many did not explain why the sample size was chosen, nor discuss sample representativeness. The heterogeneity of outcome measurement inhibited compatibility (e.g., physical health condition, injury, or psychological problem). Some of the studies gave limited information on study instruments and findings were mostly qualitatively described with no clear description of questionnaire validation. Information on socio-demographic and other associated factors is also limited in most of the studies. The reliability and comparability of primary studies were also limited by the methods of data collection and instruments used to assess occupational health hazards. Only two of the studies used specific tools to assess psychological suffering, whereas five other studies used interview schedules, FGDs, KIIs, and observation methods. Thus, the validation of instruments to assess psychological problems among child waste workers is needed to support the conduct of more rigorous and comparable studies in the future.

## 6. Implication for Research

This study revealed that there is a lack of evidence on child waste workers, which requires immediate investigation. To estimate the physical and psychological health hazards, and due to the significant work-related dangers, further epidemiological research should be conducted on occupational hazards among children involved in waste management (such as on sharp objects like metals, glass, burning wastes, and bites from insects and animals). Further epidemiological studies need to be conducted to determine the physical health risk among child garbage workers. On the other hand, specific population-based and community-specific interventions are needed to reach the children involved in waste management, because a child’s healthy psychological growth and feelings of self-confidence, will be harmed for the rest of his or her life if there is a lack of caring, including emotional support, reinforced by emotionally abusive behavior by the main adults in his or her life. This review also highlights the need for child-specific protective equipment for children involved in the waste management system. Because most protective equipment is designed for adults, it is not suitable for children. The use of appropriate protective equipment may help to reduce the number of occupational hazards.

## 7. Conclusions

There are numerous occupational hazards for child waste workers. Despite the fact that many children work in the waste management industry, they are frequently left out of traditional child protection and support systems, leaving them more vulnerable to workplace harassment and injury. The following concerns should be taken into account when creating specific programs for these vulnerable groups. To protect them from injuries and illness, they need protective gear such as facemasks, gloves, footwear, and rag-sorting tools. Access to health care and basic nutrients should be provided to them. Furthermore, children should also have access to education in order to improve their development. Finally, the authorities should also consider alternative income-generating programs for this population of children.

## Figures and Tables

**Figure 1 ijerph-19-08628-f001:**
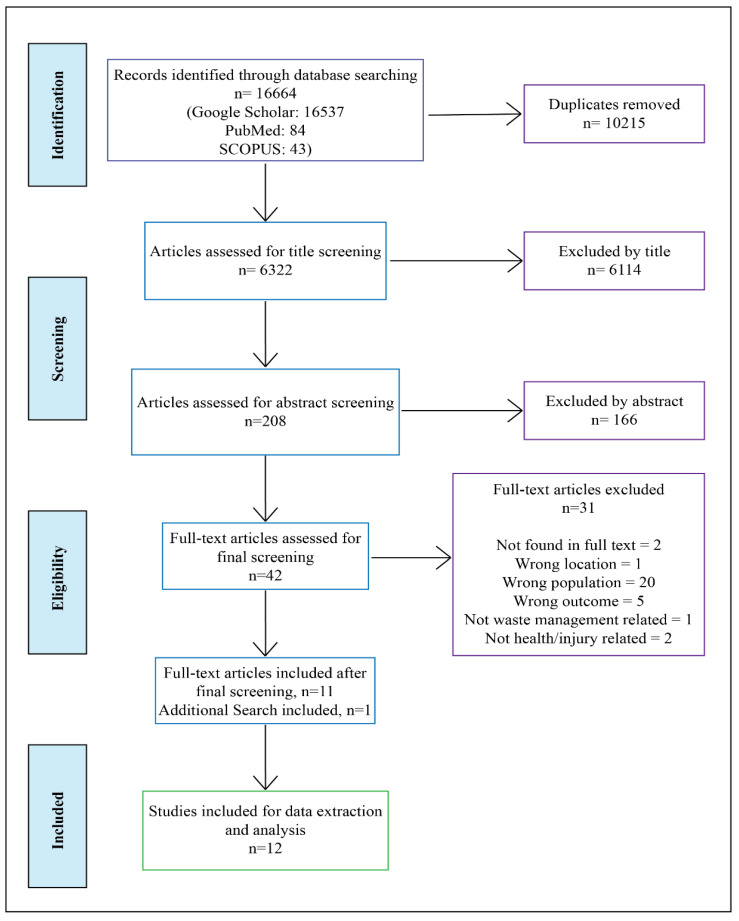
PRISMA flow diagram for searching and selecting journal articles.

**Figure 2 ijerph-19-08628-f002:**
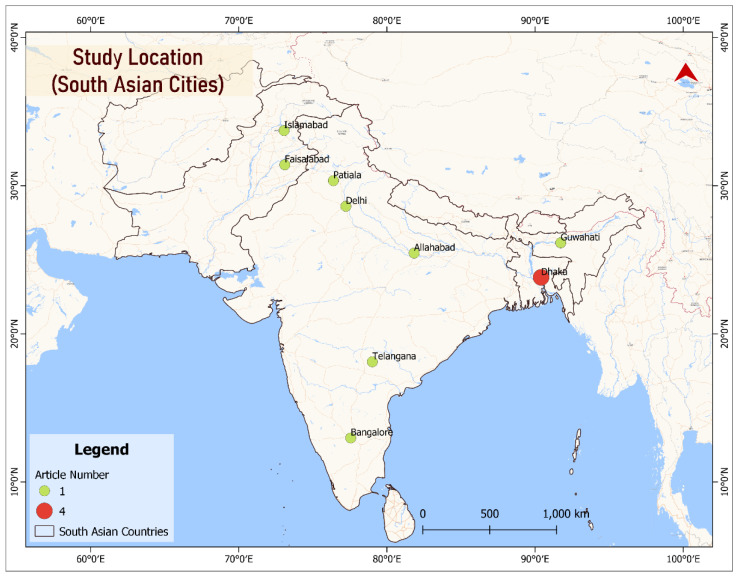
Study location with number of articles.

**Table 1 ijerph-19-08628-t001:** Description of Inclusion and Exclusion Criteria (Modified PICOTS method).

Criteria	Inclusion	Exclusion
**Population**	The children who are involved in the waste management sector and aged between 5–18 years.	The target population age is more than 18 or does not work in the waste management sector
**Relevance**	Journal articles published in the English language	Any literature published other than in English language or grey literature, including books, book chapters, conference proceedings, thesis, or articles that are not published in Scholarly journals
**Control**	Not restricted	Not applicable
**Outcome**	Health status, Occupational injury, Health suffering, Health problems	Not discussed workers’ health status, occupational injuries, or health suffering
**Time**	Journal articles published between 2000 and 2021 are included	Article published before 2000 or after 2021
**Setting/Location**	South Asian countries, Bangladesh, India, Pakistan, Nepal, Bhutan, Sri Lanka, Maldives, Afghanistan, and South Asia as a whole area included	Any country except south Asian countries

**Table 2 ijerph-19-08628-t002:** Scoping review search terms.

Issue	Search Terms
**Population Terms**	Child; Children; young; adolescent; teen; youth
**Context**	Waste collector; Waste picker; Ragpicker; informal waste picker; Waste handling worker; Garbage collector; Waste handler; Waste dumping site; Waste disposal site; Incinerator; informal waste collector; informal waste recycler; Waste recycler; Scavenger; Landfill
**Outcomes**	Health; Health impact; Health hazard; Health outcome; Health problem; Health effect; Occupational accident; Occupational injury; work accidents; Occupational Health; Health risk; work diseases; occupational diseases; work injury; Health impairment; Occupational injuries; Occupational hazards; work-related injury; work-related injuries; Health condition; work-related accident; work-related diseases; Occupational Health risk
**Location/Country**	Afghanistan; Bangladesh; India; Nepal; Bhutan; Sri Lanka; Maldives; Pakistan; South Asia

**Table 3 ijerph-19-08628-t003:** Characteristics of included studies.

Author, Year	Location	Study Design	Sample Size (R = Respondents, M = Male, F = Female)	Sampling Design and (*Study Period*)	Exposure Measurement	Outcome Variable	Quality Appraisal (Out of 10)
1. (Alam et al., 2021) [14]	Dhaka, Bangladesh	Mixed Method	R: M and F Age: 8–15 years *n* = 50	Waste collector children who worked for at least 6 months at the Matuail landfill site. Total 74 children matched the criteria and 50 participated in the survey *(September to November 2013)*	Semi-structured Questionnaire socio-demographic, health problems, treatment seeking behavior; KII (Key Informant Interview) and IDI (In-depth Interview): understanding the situation	Physical Health Sufferings, Occupational Injury	7/10
2. (Andalib et al., 2011) [15]	Dhaka, Bangladesh	Cross- sectional study	R: M and F Age: Adolescent *n* = 360	Participants were selected from four wards out of 90 in the Dhaka City Corporation area. From each area, 90 waste pickers were taken by the purposive sampling technique. *(May through August 2010)*	Semi-structured Questionnaire: socio-demographic and socioeconomic background/health problems/treatment seeking behavior	Physical Health Sufferings, Occupational Injury	5.5/10
3. (Bala & Singh, 2017) [16]	Patiala, Punjab, India	Cross- sectional study	R: M and F Age: 6–14 years *n* = 150	Rag pickers were chosen through systematic random sampling. The first respondent was chosen by lottery method. Thirty child rag pickers from each of the five Tehsils in the district were contacted *(Study period was not mentioned)*	Self-reported Questionnaire SRQ 20 (WHO): prevalence of MPD; Self-designed interview schedule: Socioeconomic and Demographic profile	Psychological Sufferings, Occupational Injury	9.5/10
4. (Batool et al., 2015) [17]	Faisalabad, Pakistan	Cross- sectional study	R: M and F Age: 7–18 years *n* = 250	The snowball sampling technique was used to select the final sample unit (rag pickers) *(Study period was not mentioned)*	Structured Questionnaire: Demographic profile/Health status/Treatment seeking	Physical Health Sufferings, Occupational Injury	6/10
5. (Dhruvarajan & Arkanath, 2000) [18]	Bangalore, Karnataka, India	Case-control study	R: F Age: 6–15 years *n* = 70 (Sample group, *n* = 35; control group, *n* = 35)	Only 1 slum was selected to minimize data biasness. The control group was of non-waste picker slum dwelling children *(Study period was not mentioned)*	Questionnaire survey: Demographic characteristic/Nutritional status/Health history and current health status; Medical examination by physicians	Physical Health Sufferings	7/10
6. (Hussian & Sharma, 2016) [19]	Delhi, India	Case-control study	R: M Age: 12–18 *n* = 120 (Rag pickers, *n* = 60; non-rag pickers, *n* = 60)	The rag pickers and non-rag pickers were randomly taken from their shelter homes and slums near shelter homes. Every third slum was identified first, and adolescents of similar age but dependent on their parents were contacted individually *(Study period was not mentioned)*	WHO (1998) Quality Of Life-BREF (Shorter Version) questionnaire and The Beck Hopelessness Scale (1993)	Psychological Sufferings	8.5/10
7. (Lahiry et al., 2011) [20]	Dhaka, Bangladesh	Case-control study	R: Not specified Age: 8–15 years *n* = 35; (Exposed group, *n* = 20; Control group, *n* = 15)	The control group was selected of the same age as subjects exposed to dump garbage, from 6 months to 6 years. *(Study period was not mentioned)*	Clinical Assessment: Blood sample tests for Oxidative stress marker/DNA damage/Liver function	Physical Health Sufferings	7/10
8. (Lal, 2019) [21]	Telangana State, India	Cross- sectional study	R: M and F Age: 5–15 years *n* = 250	Data was collected from five municipalities of the state and adopted a cluster sampling method. *(Study period was not mentioned)*	Survey and Secondary data (Details not reported)	Physical Health Sufferings, Psychological Sufferings	5/10
9. (Majumder & Rajvanshi, 2017) [22]	Allahabad, Uttar Pradesh, India	Mixed method	R: M and F Age: less than 17 years Child rag pickers, *n* = 25	Sampling method was not mentioned. *(Field survey conducted during 2016)*	Questionnaire Survey, Observations, Informal Interviews, FGD (Focus Group Discussion)	Physical Health Sufferings, Occupational Injury	5.5/10
10. Parveen & Faisal, 2005) [23]	Dhaka, Bangladesh	Case-control study	R: M and F Age: 6–15 years *n* = 150; (exposed group, *n* = 75; Control group, *n* = 75)	Stratified random sampling method *(Study period was not mentioned)*	Structured Questionnaire Survey, Interview, Physical examination. Point and Period prevalence method was used	Physical Health Sufferings, Occupational Injury	8/10
11. (Salam, 2013) [24]	Guwahati, Assam, India	Mixed method	R: M and F Age: 9–14 years *n* = 140	The sample children were selected through the purposive and snowball sampling technique. *(Study period was not mentioned)*	Interview Schedule: Demographic/Economic/Migration/Hazards/Education	Physical Health Sufferings, Occupational Injury, Psychological Sufferings	6/10
12. (Shehzad, Jalal 2014) [25]	Islamabad, Pakistan	Qualitative	R: M and F Age: less than 18 years *n* = 50	Respondents were selected through purposive sampling. *(Study period was not mentioned)*	Interview Schedule: Violence in the home, school, workplace, and society	Physical Health Sufferings, Psychological Sufferings	6.5/10

**Table 4 ijerph-19-08628-t004:** Occupational injuries among child waste workers.

Author, Year	Types of Occupational Injuries	Frequency of Injuries	
		Child Waste Worker	Control Group
1. (Alam et al., 2021) [14]	Cuts	50.0%	-
	Bruise	6.0%	-
	Stray Animals	8.0%	-
	Rats/mice	8.0%	-
2. (Andalib et al., 2011) [15]	Injuries	59.4%	-
	Cuts in hand	52.9% (of injured)	-
	Injury in legs	25.0% (of injured)	-
	Other types of injury	22.1% (of injured)	-
3. (Bala & Singh, 2017) [16]	Injured during work	93.3%	-
4. (Batool et al., 2015) [17]	Injuries	62.8%	-
	Animal bite	2.8%	-
	Bruise	6.4%	-
	Cuts	53.6%	-
	Contact with Feces and stray animals	68.8%	-
	Contact with Chemical fumes	11.2%	-
	Contact with Airborne dust	28.0%	-
5. (Lal, 2019) [21]	Rashes, cuts, wounds	58%	-
	Dog and snakebite	24%	-
6. (Majumder & Rajvanshi, 2017) [22]	Bitten by animals and insects	92.0%	-
	Cuts by broken glass	84.0%	-
7. (Parveen & Faisal, 2005) [23]	Wound/Injury problems	80.3%	6.7%
	Cuts from sharp objects	80.3%	6.7%
	Injury caused by machines	1.0%	0.0%
8. (Salam, 2013) [24]	Injuries	70.0%	-
	Accidents	16.4%	-
	Dog bite	30.7%	-
	Insect bite	25.7%	-
	Poisonous Weeds	5.0%	-

**Table 5 ijerph-19-08628-t005:** Physical health suffering of child waste workers.

Author, Year	Types of Physical Health Suffering	Frequency of Sufferings		Statistical Significance
		Child Waste Workers	Control Group	
1. (Alam et al., 2021) [14]	Fever	64.0%	-	-
	Fatigue	68.0%	-	-
	Dizziness	86.0%	-	-
	Arthritis (Joint pain)	14.0%	-	-
	Back Pain	30.0%	-	-
	Bone fracture	2.0%	-	-
	Skin rash	12.0%	-	-
	Dermatitis	2.0%	-	-
	Scabies	52.0%	-	-
	Cough	68.0%	-	-
	Hemoptysis (coughing with blood)	2.0%	-	-
	Dyspnea (shortness of breath)	18.0%	-	-
	Abdominal Pain	36.0%	-	-
	Diarrhea	10.0%	-	-
	Vomiting	30.0%	-	-
	Eye vision problems	6.0%	-	-
	Dental problems	16.0%	-	-
	Parasites (i.e., worms)	4.0%	-	-
	Head lice	2.0%	-	-
	Growth retardation	64.0%	-	-
2. (Andalib et al., 2011) [15]	Suffered in the last 6 months from any health problems	80.0%	-	-
	Skin disease	31.2%	-	-
	Cough	21.9%	-	-
	Fever	20.1%	-	-
	Diarrhea	19.1%	-	-
	Itching	11.1%	-	-
	Jaundice	3.1%	-	-
	Anemia	51.5%	-	-
	Eczema	19.7%	-	-
	Scabies	14.4%	-	-
	Wounds	15.2%	-	-
	Common Cold	1.4%	-	-
	Goiter	10.6%	-	-
	Todd skin	6.8%	-	-
3. (Batool et al., 2015) [17]	Suffering from diseases within last one year	67.2%	-	-
	Digestion Problems	15.6%	-	-
	Skin problems	13.6%	-	-
	Respiratory Problems	9.2%	-	-
	Back and Joint pain	6.8%	-	-
	Cough	6.4%	-	-
	Headache	3.6%	-	-
	Fever	2.8%	-	-
	Tuberculosis	0.8%	-	-
4. (Dhruvarajan & Arkanath, 2000) [18]	Acute and Intermittent fever	5.7%	5.7%	-
	Acute and Continuous fever	11.4%	20%	-
	Chronic and Continuous fever	8.6%	0%	-
	Non-Productive Cough	8.6%	0%	-
	Productive Cough	25.7%	34.3%	-
	Productive Cough (Yellow Sputum—Indicates infection)	8.6%	22.9%	-
	Productive Cough (White Sputum—Indicates chronic condition)	17.1%	11.1%	-
	Respiratory Problem	62.8%	40%	z = 3.80 (Significant at 1%)
	Gastrointestinal Problem	34.3%	8.6%	z = 4.36 (Significant at 1%)
	Worm Infestation	8.6%	8.6%	z = 0 (not Significant)
	Skin Problem	5.7%	2.8%	z = 2.99 (Significant at 1%)
5. (Lahiry et al., 2011) [20]	[Oxidative stress induced damage] Lipid hydroperoxide (nmol mL^−1^)	12.21 ± 4.98 ** (Mean + SE)	7.63 ± 0.38 (Mean + SE)	** *p* < 0.01
	[Oxidative stress induced damage] TBARS value (nmol MDA eq mL^−1^)	15.99 ± 4.61 *** (Mean + SE)	6.37 ± 0.41 (Mean + SE)	*** *p* < 0.001
	[Oxidative stress induced damage] Protein carbonyl value (nmol mg^−1^ of protein)	951.58 ± 154.6 ** (Mean + SE)	394.74 ± 25.56 (Mean + SE)	** *p* < 0.01
	[DNA damage] Head DNA (%)	71.76 ± 1.78 *** (Mean + SEM)	95.23 ± 1.57 (Mean + SEM)	*** *p* < 0.001
	[DNA damage] Tail DNA (%)	28.24 ± 1.07 *** (Mean + SEM)	4.77 ± 1.09 (Mean + SEM)	*** *p* < 0.001
	[DNA damage] Tail Moment (%)	5.93 ± 0.19 *** (Mean + SEM)	0.38 ± 0.01 (Mean + SEM)	*** *p* < 0.001
	Serum bilirubin	0.95 ± 0.12 (Mean + SEM)	0.62 ± 0.09 (Mean + SEM)	Not significant
	Serum albumin	4.15 ± 0.69 (Mean + SEM)	4.35 ± 0.81 (Mean + SEM)	Not significant
6. (Lal, 2019) [21]	Total Physical Hazards *	50%	-	-
	Total Biological Hazards *	17%	-	-
7. (Majumder & Rajvanshi, 2017) [22]	Suffered from fever, cold, tetanus, skin problem, headache, pain in bone joints, eye infections, and backache.			
	Children felt fatigued working for long hours.			
	Consuming rotten food items from garbage bins and eating without washing hands led to stomachaches.			
	Rag-picking children had less resistance to diseases because of malnutrition.			
8. Parveen & Faisal, 2005) [23]	**General Health problems**	97.3%	70.7%	-
	Weakness	96.0%	49.3%	-
	Dizziness	90.7%	9.3%	-
	Loss of appetite	88.0%	36.0%	-
	Burning sensation	88.0%	10.7%	-
	Swelling limbs	32.0%	1.3%	-
	UTI	42.7%	8.0%	-
	**Ache and Pain problems**	94.7%	69.3%	-
	Headache	93.3%	57.3%	-
	Back pain	82.7%	13.3%	-
	Pain in the joint	82.7%	12.0%	-
	**Skin problems**	97.3%	58.7%	-
	Itching	73.3%	25.3%	-
	Eczema	24.0%	2.7%	-
	Scabies	38.7%	18.7%	-
	Abscess	56.0%	0.0%	-
	Lice	81.3%	32.0%	-
	**Respiratory problems**	85.3%	46.7%	-
	Cough	82.7%	42.7%	-
	Breathing Problem	60.0%	6.7%	-
	Blood with cough	4.0%	2.7%	-
	Throat Infection	46.7%	6.7%	-
	Chest Pain	53.3%	4.0%	-
	Oral infection	49.3%	1.3%	-
	**Gastrointestinal problems**	85.3%	74.7%	-
	Acidity	81.3%	32.0%	-
	Loose motion and Vomiting	76.0%	37.3%	-
	Blood Dysentery	64.0%	4.0%	-
	Pain in stomach	76.0%	29.3%	-
	**Eye problems**	65.3%	10.7%	-
	Eye irritation	53.3%	10.7%	-
	Blurry vision	25.3%	1.3%	-
	Eye infection	22.7%	1.3%	-
	Night blindness	16.0%	0.0%	-
	**Fever Problems**	85.3%	22.7%	-
	Fever	70.7%	13.3%	-
	Fever blister	4.0%	0.0%	-
	Persistent fever	9.3%	8.0%	-
	Recurrent fever	32.0%	1.3%	-
	**Ear problems**	33.3%	16.0%	-
	Ear pain	24.0%	10.7%	-
	Ear infection	29.3%	6.7%	-
	Loss of hearing	1.3%	2.7%	-
9. (Salam, 2013) [24]	Diseases *	42.1%	-	-
	Fatigue	71.4%	-	-
10. (Shehzad, Jalal 2014) [25]	Physical violence within the family context	56.0%	-	-
	Sexual violence within the family context	4.0%	-	-
	Physical violence at educational institutions	10.0%	-	-
	Physical violence in the workplace	72.0%	-	-
	Physical violence in the community	66.0%	-	-
	Sexual violence in the community	6.0%	-	-

* Details not provided. ** *p* < 0.01, *** *p* < 0.001

**Table 6 ijerph-19-08628-t006:** Psychological suffering of child waste workers.

Author, Year	Types of Psychological Suffering	Frequency of Psychological Suffering		Statistical Significance
		Child Waste Workers	Control	
1. (Alam et al., 2021) [14]	Developmental/mental retardation	68.0%		
2. (Bala & Singh, 2017) [16]	Overall prevalence of MPD	42.7%	-	
	MPD prevalence among Boys	39.8%	-	*p*-value 0.045
	MPD prevalence among Girls	53.1%	-	
	MPD prevalence among Physically injured	45.0%	-	*p*-value 0.001
	MPD prevalence among not injured	10.0%	-	
	MPD prevalence among Low-income group (Less than 2000 rupees/month)	50.7%	-	*p*-value 0.003
	MPD prevalence among High-income group (2000 to 4000 rupees/month)	35.2%	-	
	MPD prevalence among Smokers	49.2%	-	*p*-value 0.012
	MPD prevalence among Non-smokers	37.6%	-	
	MPD prevalence among Pan (Betel leaf) consumers	44.2%	-	*p*-value 0.001
	MPD prevalence among Non-pan (Betel leaf) consumers	40.0%	-	
	MPD prevalence among children satisfied with the job	39.3%	-	*p*-value 0.001
	MPD prevalence among children not satisfied with the job	51.2%	-	
3. (Hussian & Sharma, 2016) [19]	Physical health *	14.8 (Mean)	18.7 (Mean)	*p*-value 0.103; t-value 3.42
	Psychological Health*	13.0 (Mean)	17.7 (Mean)	*p*-value 0.055; t-value 3.82
	Quality of Life Overall score	50.2 (Mean)	67.2 (Mean)	*p*-value 0.001; t-value 4.18
	Hopelessness	14.4 (Mean)	10.0 (Mean)	*p*-value 0.047; t-value 4.49
	Correlation coefficient between Quality of Life and Hopelessness	−0.8 (r)	−0.1 (r)	-
4. (Lal, 2019) [21]	Total Psychological hazards *	18.0%	-	-
5. (Salam, 2013) [24]	Feeling of insult	12.9%	-	-
	Fear of people	7.1%	-	-
6. (Shehzad, Jalal 2014) [25]	Psychological violence within family context (Insult, threats, isolation, rejection)	100.0%	-	-
	Psychological violence at educational institutions	20.0%	-	-
	Psychological violence in the workplace	92.0%	-	-
	Psychological violence in the community	84.0%	-	-

* Details not provided.

**Table 7 ijerph-19-08628-t007:** Healthcare seeking behavior among child waste workers.

Author, Year	Healthcare Seeking Behavior	Frequency (%)
1. (Alam et al., 2021) [14]	Did not seek treatment	6 (12.0%)
	NGO	3 (6.0%)
	Pharmacy	20 (40.0%)
	Community Hospital	1 (2.0%)
	Health Centre	2 (4.0%)
	Government Hospital	1 (2.0%)
	Free Medical Camp	1 (2.0%)
	Private Hospital	2 (4.0%)
	Government Hospital, Pharmacy	5 (10.0%)
	Free camp, Pharmacy	3 (6.0%)
	NGO, Pharmacy	3 (6.0%)
	NGO, Private Hospital	1 (2.0%)
	NGO, Health Officers	2 (4.0%)
2. (Andalib et al., 2011) [15]	Did not seek treatment	114 (39.6%)
	Homeopath	13 (4.5%)
	Traditional Healer	33 (11.5%)
	Medicine Seller	128 (44.4%)
3. (Batool et al., 2015) [17]	Consulted the local doctors	149 (59.6%)
	Did not consult with anyone (self-treatment)	57 (22.8%)
	Hospital	20 (8.0%)
	Homeopathic	13 (5.2%)
	Hakeem	11 (4.4)
4. (Lal, B Suresh, 2019) [21]	Seek treatment	(74.0%)
5. (Majumder & Rajvanshi, 2017) [22]	Took medical treatment (medical practitioners, medical shops)	(24.0%)
	Took treatment at home, and zadu-tona (black magic)	(76.0%)

## Data Availability

Not applicable.

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
