# Peer review of "Occupational Health Sufferings of Child Waste Workers in South Asia: A Scoping Review"

_ijerph, 2022, doi:10.3390/ijerph19148628_

Round 1

Reviewer 1 Report

Note: the link to the paper no longer works so I am unable to access it to write this critique. I have printed the body of the paper only so I am unable to comment on the references or table.

General: The paper covers a very important and concerning topic, and the authors used a robust method to perform the literature review. However, the English grammar, sentence construction, and flow of the content is not of sufficient quality to warrant publication. Much work needs to be done to write it so English readers can understand it properly. Someone whose first language is English who also has experience with scientific writing should work with the authors to rewrite this paper.

Specific comments:

2.3 The paper refers to “keywords” and “search terms” which are the same thing. Choose one phrase for the manuscript, not both.

The exclusion criteria states populations more than 17 yrs old are excluded, yet the data extraction table and the body of the article itself includes much information that pertains to adults. The results and discussion sections do not make clear which data pertains to children and includes data about adults, which is not the aim of the paper. This needs to be made much clearer throughout the paper. For example, section 3.2 – do these findings all pertain to children specifically?

Use of the terms “etc” and “and so on” should not be used in a scientific paper since they do not describe data points accurately.

3.3 "Children suffer from skin problems as a result of their exposure to waste." How do you know there is a causal link? Might not there be other possible reasons for skin problems? These causal connections cannot be assumed in a scientific paper.

3.4 "Psychological suffering is the root cause of social unrest": this sentence needs a reference or else it should be removed.

The discussion section contains much information that belongs in the results section. Discussion section should not include results that should appear in the Results section; it should include comparisons to other similar studies (in this case, child waste workers in other parts of the world).

Discussion section, first paragraph has several statements that need references.

4.2 paragraph 4: How is it possible that women were not included in the studies? In the background section of the paper it is noted that women make up many waste collectors. Moreover, this paper is supposed to be about children, not women.

4.3 The term “microbial load” is used. This needs to be explained. This is not a clinical measurement used in Western medicine.

Reviewer 2 Report

Occupational Health Sufferings of Child Waste Workers in South Asia: A Scoping Review

This is a very interesting attempt to assess child labor from an occupational health perspective in South Asian countries in a quantitative and qualitative approach. The authors have made a significant effort to identify relevant publications on this important topic. It is a well written study with rich information on different aspects of the issue.

General comments:

  1. The data extraction table may be better to be presented as an appendix.
  2. The use of the word percent after a numerical should be consistent in the whole text. Either report a number with the word percent next or report a number with the sign % next to the number
  3. Never begin a sentence with a numerical. You may begin the sentence with “A total of…”
  4. Use one decimal point for all numbers reported. Others have no decimal points. Others are reported with two decimal points. Be consistent with one decimal point.
  5. When reporting percentages and findings from different studies, sometimes the authors say the percent was… and in other occasions they say the percentage is… Be consistent by reporting the all numbers as found in the past (the percentage was…)
  6. Limitations section goes last or reported right before the conclusions?

Specific comments:

  1. It would be helpful for the reader to see one compact solid Table with the different occupational injuries of children and the range of percentages found in the different studies. Reporting all of these numbers in the text of the results is not readily available for the reader.
  2. The limitations sections should be elaborated with points related to the methodological approach of the study. For example, why they authors were not able to identify any study from so many countries (Nepal, Bhutan, Sri Lanka, Maldives, Afghanistan. Was there any pitfall in their search strategy?
  3. Did the authors try to search manually to identify studies on their subject from the countries not represented in their results?
  4. Why there was such a huge asymmetry in the number of studies identified from Google Scholar as compared to PubMed and SCOPUS?
  5. The authors should also report the final number of studies from Google Scholar, PubMed, and SCOPUS included in their results sections, respectively.

Editorial comments:

  1. Pages starting from 16 onward … are not numbered.
  2. Line 21 correct to “Respiratory”
  3. Line 61 delete the word “conditions”
  4. Line 117 delete the word “titled” and correct the next word “was” to “were”
  5. Line 118 insert a full stop after the parenthesis. Continue with Capital “These include”
  6. Line 150 replace the word comprehending with “Exploring”
  7. Line 161 replace the words women and men with “boys” and “girls”
  8. Bottom of page 20 correct the font for “communicable diseases”
  9. Middle of page 21 (3rd paragraph). 1st line of 3rd Hopelessness … Needs a syntax correction.

Author Response

Please see the attachements

Round 2

Reviewer 1 Report

The paper is much improved and should be published. The term "and so on" still appears at least once and is not acceptable when describing study data. Use of 2 decimal points is not necessary when listing prevalence of health effects. There are still multiple sentence fragments - missing either a verb, subject, or predicate - which should be fixed before publication. 

Author Response

We appreciate your valuable and thorough assessment. The term "and so on" was deleted. Additionally, one decimal is used in the result and comment sections. We make every effort to resolve all other problems.

Reviewer 2 Report

The authors have made a very good effort to revise their manuscript accordingly. There are still some typos. Authors need to go through their manuscript once more to correct typos and provide a careful editorial review before finalizing their article.

Author Response

We greatly appreciate your insightful comments and extensive reviews. We have made an effort to carefully read the entire content and fix any errors.